# Emergence of cooperative bistability and robustness of gene regulatory networks

**Shintaro Nagata** [1,2☯], **Macoto Kikuchi** [1,2,3☯] *

**1** Department of Physics, Osaka University, Toyonaka, Japan, **2** Cybermedia Center, Osaka University, Toyonaka, Japan, **3** Graduate School of Frontier Bioscience, Osaka University, Suita, Japan

☯ These authors contributed equally to this work.
* kikuchi@cmc.osaka-u.ac.jp

**Data Availability Statement:** All data files and source codes are available at Zenodo repository (DOI: 10.5281/zenodo.3716026).

**Funding:** This work was supported by JPSJ KAKENHI Grant Number 15K05246. The funders

## Abstract

Gene regulatory networks (GRNs) are complex systems in which many genes regulate mutually to adapt the cell state to environmental conditions. In addition to function, the GRNs possess several kinds of robustness. This robustness means that systems do not lose their functionality when exposed to disturbances such as mutations or noise, and is widely observed at many levels in living systems. Both function and robustness have been acquired through evolution. In this respect, GRNs utilized in living systems are rare among all possible GRNs. In this study, we explored the fitness landscape of GRNs and investigated how robustness emerged in highly-fit GRNs. We considered a toy model of GRNs with one input gene and one output gene. The difference in the expression level of the output gene between two input states, "on" and "off", was considered as fitness. Thus, the determination of the fitness of a GRN was based on how sensitively it responded to the input. We employed the multicanonical Monte Carlo method, which can sample GRNs randomly in a wide range of fitness levels, and classified the GRNs according to their fitness. As a result, the following properties were found: (1) Highly-fit GRNs exhibited bistability for intermediate input between "on" and "off". This means that such GRNs responded to two input states by using different fixed points of dynamics. This bistability emerges necessarily as fitness increases. (2) These highly-fit GRNs were robust against noise because of their bistability. In other words, noise robustness is a byproduct of high fitness. (3) GRNs that were robust against mutations were not extremely rare among the highly-fit GRNs. This implies that mutational robustness is readily acquired through the evolutionary process. These properties are universal irrespective of the evolutionary pathway, because the results do not rely on evolutionary simulation.

## Author summary

Living systems have developed through a long history of Darwinian evolution. They acquired characteristic properties distinct from other physical systems; one is biological function. Another important property, which is overlooked by non-experts, is robustness to noise and mutation. Here, robustness means that a system does not lose its

had no role in study design, data collection and analysis, decision to publish, or preparation of the manuscript.

**Competing interests:** The authors have declared that no competing interests exist.

functionality when exposed to disturbances. Then, how do they relate to each other? In this paper, we explored this question using a toy model of gene regulatory networks (GRNs). While evolutionary simulations are usually used for such purposes, we instead generated GRNs randomly and classified them according to functionality. By requiring sensitive responses to environmental change as a function, we found that bistability emerges as a common property of highly-functional GRNs. Since this property does not depend on a particular evolutionary pathway, if the evolution was rewound and repeated over and over again, phenotypes with the same property would always evolve. At the same time, such bistable GRNs were robust to noise. We also found that GRNs robust to mutation were not extremely rare among the highly-functional GRNs. This implies that mutational robustness would be readily acquired through evolution.

## Introduction

Living systems have advanced through a long history of Darwinian evolution. As a result, they have acquired properties distinct from other physical systems. First, they have developed biological functions. For example, at the molecular level, many different proteins have evolved to work under physiological conditions. On a larger scale, the cells have developed to undergo metabolism and proliferation.

Another significant property of living systems is the existence of several types of robustness [1–3]. Here, the term robustness refers to the property of a system retaining its function despite exposure to disturbances such as mutations or noise. Among the types of robustness, mutational robustness is of particular importance. Since living systems are constantly exposed to the mutation of genes, the systems that are not robust against mutation do not survive the course of evolution. Thus, it is widely considered that mutational robustness is acquired through evolution. However, if we regard the process of developing functions as an optimization process, highly optimized systems are intuitively considered as fragile against disturbances, and thus, it is natural to consider that they readily lose their functions by mutation. In this respect, the process of evolution should be something different from a simple optimization process. Robustness against several types of noise, such as disturbance from the environment and the noise that occurs within the cell, and in particular, the fluctuation due to the finiteness of the number of molecules, is also important, because living systems function stochastically in the noisy world in which we live.

As Wagner pointed out [2], it is difficult to investigate mutational robustness experimentally. Thus, numerical simulations based on mathematical models provide important information. In this work, we investigated the relationship between the development of function and robustness using a toy model of gene regulatory networks (GRNs).

GRNs are complex networks of genes that mutually regulate their expression levels to adapt the cell state to environmental conditions. The regulation mechanism of GRNs is as follows: A protein called a transcription factor (TF) is expressed from one gene. This TF acts either as the activator or the repressor of other genes. The activator binds to an upstream region of the promoter of the target gene and promotes its expression by recruiting the RNA polymerase, which produces mRNA. A protein is synthesized at a ribosome according to the mRNA sequence. A gene is sometimes expressed without the activator owing to spontaneous binding of the RNA polymerase. In contrast, the repressor binds to the promoter of the target gene to block RNA polymerase. As a result, the expression of the target gene is prevented. The produced protein then acts as a TF to yet other genes, and thus the genes regulate each other.

There have been a number of theoretical studies on the evolution of GRNs, most of which have used evolutionary simulations [4–11]. However, the results and knowledge obtained from these types of studies strongly depend on the evolutionary pathways studied. We discuss the universal aspects of function and robustness, irrespective of the evolutionary pathway, by exploring a landscape description of evolution. Landscape descriptions have been discussed in biological systems in many different contexts. Some examples include the energy landscape for protein folding [12] and the epigenetic landscape for development [13]. The fitness landscape (or adaptiveness landscape), in which the fitness distribution in the multidimensional genotypic space is considered, is a classic concept for discussing evolutionary processes [14]. However, it has long been viewed as an abstract concept, except for when applied to very simple models. For example, the ruggedness of the fitness landscape has been discussed using the random Boolean network model [15–17]. Quite recently, empirical fitness landscapes are becoming available [18].

We explored a different type of a fitness landscape, in which the fitness was considered as an independent variable. We classified GRNs according to their fitness and investigated the properties that emerged as the fitness increased. Research in this direction was previously conducted by Ciliberti *et al.* [19, 20], who sampled GRNs randomly and found that a majority of functional GRNs are connected with each other by successive mutations, similar to the neutral networks found in the RNA sequence space [21]. In other words, the functional GRNs form a large cluster in a neutral space, which is the genotypic populations that share the same fitness [2]. For such genotypes, the possibility that they stay in the same neutral space is high when some of the genes are mutated. Thus, the genotypes belonging to such a cluster are considered to be robust against mutations. The concept of the neutral space is also a landscape-type point of view. It should be noted, however, that the fitness of the model used by Ref. [19] had only two values: either viable or not viable. This simplification made it possible to investigate the neutral space by randomly sampling GRNs.

The model we constructed has a continuous fitness. We considered random GRNs, which have one input gene and one output gene, like the one used by Ref. [10]. They can be regarded either as GRNs directly responding to environmental changes or as a part of a larger GRN. We assumed that the input signal took two distinct states ("on" and "off" for example) and required that the response to these two input states differ as largely as possible. In other words, the required function of GRNs was to distinguish between the two input states. We defined the difference in the expression of the output gene between two input states as the fitness. We constructed ensembles of GRNs in which possible GRNs are classified according to their fitness and investigated how the characteristic properties of GRNs change with the fitness. The constant-fitness ensemble introduced in this study is a concept very similar to the neutral space. If evolution is the process during which the fitness is gradually optimized, it should consist of successive transitions between the constant-fitness ensembles in the direction of a higher fitness. Thus, if some universal properties emerge in this process, they should be observed, irrespective of the evolutionary pathway.

Since GRNs having high fitness are rare, the simple random sampling method is not appropriate for obtaining a sufficient number of samples of such GRNs. In studies by Refs. [22, 23], the Markov Chain Monte Carlo method (Metropolis method) was used to sample functional GRNs randomly, for which the network motifs were analyzed. This method is suitable for efficient sampling of functional GRNs. In this paper, in contrast, to sample GRNs in the whole range of fitness, we employed a rare event sampling method based on the multicanonical Monte Carlo (McMC) method, which was developed in the field of equilibrium statistical mechanics.

## Model and method

### Model

We ignored the detailed mechanism of the expression of transcription factors and dealt only with the regulatory relationships between genes. Namely, we considered a connectionism-type model [24]. A GRN is represented by a directed random graph in which the nodes correspond to the genes and the edges correspond to the regulatory interactions. To exclude genes not participating in the functions, we required that paths from the input gene to all the other genes, as well as paths to the output gene from all the other genes, exist. We prohibited both the mutual regulation of two genes and the self-regulating loop. They are frequently observed in real GRNs and in particular, the combination of mutual repression and self-activation is known to give rise to bistability [22, 25–27]. Thus, this restriction means that we excluded a representative structure of bistable GRNs. We conducted preliminary computations for the model that these two types of regulations are permitted and confirmed that the results were qualitatively unchanged. Regulations to the input gene from the other genes and regulations from the output gene to the other genes were permitted.

The GRNs have $N$ nodes and $K$ edges. The average number of edges connected to a node is represented as $C \equiv 2K/N$. The node number one was assigned to the input node. How the output node was determined will be described later. An illustrative example of a randomly-generated GRN is shown in Fig 1.

Each node was assigned a variable called "expression" $x_i$, where $i$ is the node number and $x_i$ takes a continuous value $\in [0, 1]$. We assumed the following difference equation for the dynamics of the expression as in Ref. [4, 5]:

$$x_i(t+1) = R\left( I(t)\delta_{i,1} + \sum_{j \neq i} J_{ij} x_j(t) \right), \tag{1}$$

$$R(x) = \frac{1}{1 + e^{-a(x-b)}}, \tag{2}$$

where $I(t)$ is the input signal at time $t$, which is applied only to the input node. $J_{ij}$ expresses the regulating interaction from $j$-th node to $i$-th node. For simplicity, all amplitudes of the regulations were taken as $|J_{ij}| = 1$ if they existed, and 0 or $\pm 1$ were assigned randomly to $J_{ij}$: $J_{ij} = 0$ means that there is no regulating interaction from $j$-th gene to $i$-th gene and $J_{ij} = 1$ and $-1$ represent the activation and the repression, respectively. Since self-regulation was not permitted, $J_{ii} = 0$. $J_{ji} = 0$ when $J_{ij} \neq 0$, because mutual regulation was absent.

The discontinuous sign function is sometimes used for the response function $R(x)$ with $x$ being assumed to take $\pm 1$ [4, 5, 20]. But since we want to define a fitness function having a continuous value, we used the above sigmoidal function, which reflects the fact that the responses in living systems are imprecise and stochastic (or "sloppy" according to Ref. [11]). This type of response function was proposed in Ref. [4] and was frequently used [6, 10, 11, 28, 29]. The parameter $a$ determines the steepness of the sigmoidal function and $b$ gives the threshold. In this paper, we fixed the parameters to $a = 1$ and $b = 0$. In this case, $R(x)$ is equivalent to (tanh $x$ + 1)/2 [8]. The function is also close to the one proposed by Mjorsness inspired by the Hill function [24]. Each node exhibits the spontaneous expression $x_i = 0.5$ even when no input is provided. We also conducted preliminary computations for steeper functions such as (tanh $2x$ + 1)/2, which corresponds to a response function with a larger Hill coefficient, and confirmed that results were qualitatively unchanged.

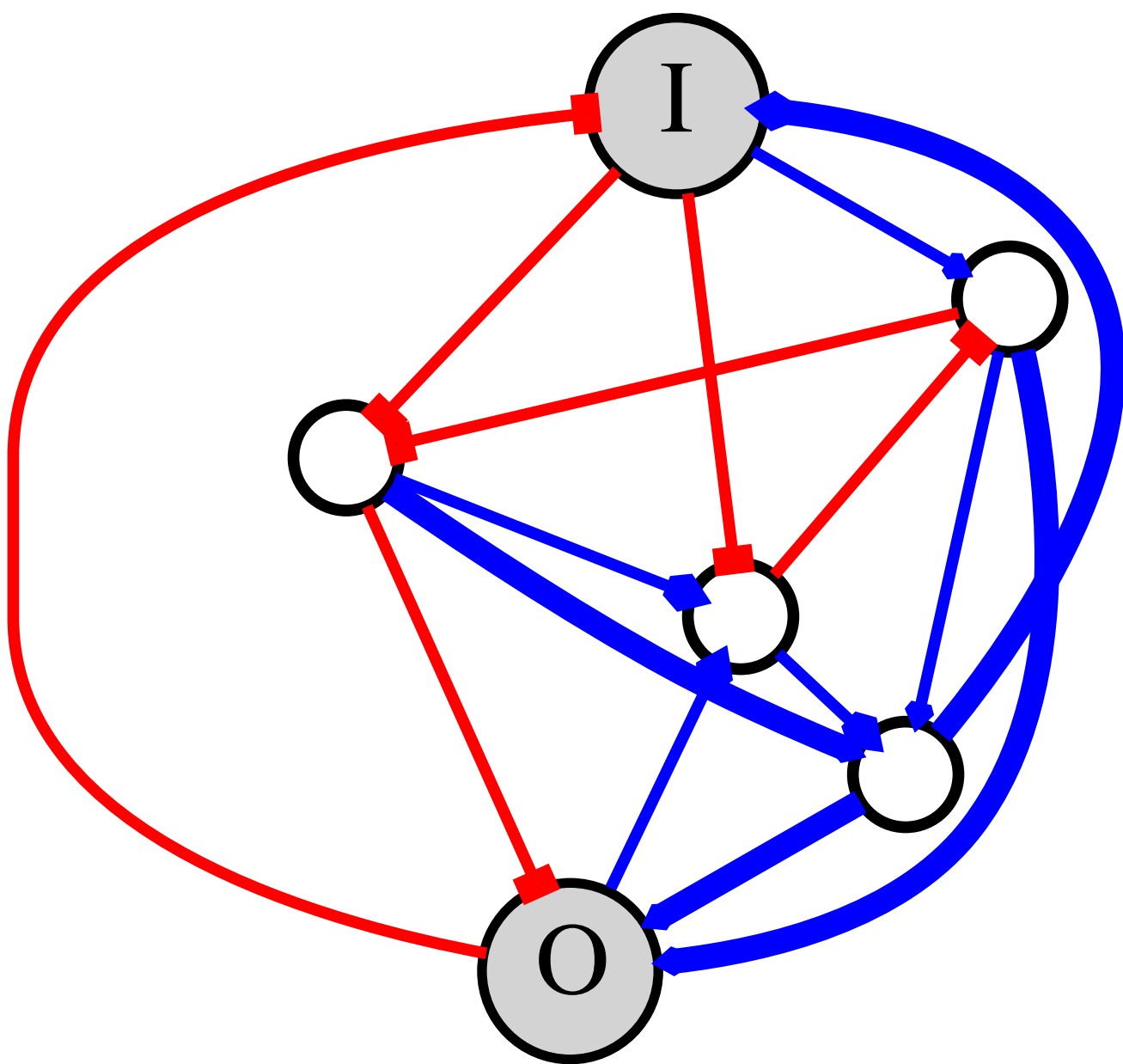

**Fig 1. An example of the random GRN.** A randomly generated GRN with one input node and one output node for $N = 6$ and $K = 15$ is shown. I and O indicate the input node and the output node, respectively. The lines indicate the regulation: The blue lines are for activation and the red lines are for repression.

Assuming that the environment takes two distinct states, we required the GRN to respond to the difference between $I = 0$ and 1 as sensitively as possible. This is the functionality assumed for GRNs in this study. For that purpose, we defined the response $\bar{x}_i(I)$ of $i$-th gene to the input as the temporal average of the expression at the steady state. If the dynamical system reached the fixed point, taking the average was not necessary. The system reached the fixed point in most cases. As the initial state, we assumed that all the genes exhibited a spontaneous expression and set $x_i = 0.5$. The initial state dependence will be discussed later on.

The sensitivity of the $i$-th gene was defined by the difference of the response of the gene for $I = 0$ and 1 as follows:

$$s_i = |\bar{x}_i(1) - \bar{x}_i(0)|. \tag{3}$$

It should be noted that we only required the absolute value of the difference between $\bar{x}_i(1)$ and $\bar{x}_i(0)$ to become large in this definition, and which value was larger was not relevant. The node that exhibited the largest sensitivity except the input node was selected as the output node. If the node of the largest sensitivity lacked paths from one or more other genes, such GRN was not used. We regarded the sensitivity of the output node as the "fitness" $f$. Since we did not perform the evolutionary simulation, the term "fitness" simply refers to the degree of functionality.

## Method: Multicanonical ensemble

Our purpose was to classify the randomly generated GRNs according to fitness and investigate their universal properties. To this end, we would have liked to have sampled a large number of GRNs with a variety of fitness values, but as GRNs with high fitness were expected to be rare, the simple random sampling procedure was considered not to be useful. Thus, we employed the rare-event sampling method based on the multicanonical Monte Carlo (McMC) method [30, 31].

McMC belongs to the Markov chain Monte Carlo method and was originally developed in the field of equilibrium statistical mechanics. In the context of statistical mechanics, it realizes the uniform sampling in energy. For that to be possible, the appearance probability of a microscopic state of energy $E$ should be made inversely proportional to the number of states $\Omega(E)$ having the same energy. In the ordinary Metropolis method, the following detailed balance condition is assumed between the transition probability $W_{ij}$ from $j$-th state to $i$-th state and that for the inverse process:

$$W_{ij}P_{eq}(E_j) = W_{ji}P_{eq}(E_i), \tag{4}$$

where $P_{eq}(E)$ represents the appearance probability of the microscopic state of the energy $E$ in thermal equilibrium. While the Gibbs distribution is used as $P_{eq}(E)$ in the ordinary Metropolis method, the flat energy distribution is obtained if $P_{eq}(E) \propto 1/\Omega(E)$.

Although the number of states $\Omega(E)$ is not known beforehand, only a rough estimation is sufficient for the sampling purpose. McMC consists of two processes. The first is the learning process to determine the weight, namely, the approximate value of the appearance probability for each energy state. The second step is the measurement process by the Metropolis method using these weights. In the case that the energy takes continuous values, the whole range of the energy is divided into bins and $\Omega(E)$ is approximated by a piecewise linear function in the original McMC. The probability distribution of the energy in each bin is regarded as the canonical distribution of constant temperature, which differs from bin to bin. In contrast, we assign a constant weight within each bin. Thus the energy distribution in each bin is the microcanonical distribution. This latter method is called "entropic sampling" [32], which is one of the variants of McMC.

By using this method, we can sample the microscopic states in a wide range of energies randomly, in principle. We can estimate the appearance probability of the energy corresponding to each bin using the obtained histogram and the weight. The energy distribution in a bin obtained by entropic sampling is not uniform but proportional to the number of states having the same energy. Thus, for thermodynamic systems, the number of states that appear in the simulation within each bin decays exponentially as the energy increases. It should be noted

that since this method is based on the Markov chain Monte Carlo method, the successive samples resemble each other. Therefore, samples should be taken at intervals to reduce these correlations.

By regarding quantities other than the physical energy as energy, this method can be applied to a variety of problems. It is particularly suitable for counting rare states or for estimating the appearance probability of rare events [33–38]. Ref. [36] used this method to investigate the mutational robustness of biologically-inspired networks.

In this study, we performed entropic sampling by regarding fitness as energy. We estimated the appearance probability of each fitness and sampled GRNs with a wide range of fitness values. As the elementary process of the Monte Carlo method, we replaced a randomly chosen edge to another place and selected the input and output nodes as the abovementioned conditions were satisfied. Thus $K$ was kept constant. It should be noted that this elementary process is only for sampling GRNs in McMC and does not relate to any evolutionary process.

For the networks having $K$ edges, we defined $K$ trials of elementary processes as one Monte Carlo step (MCS). We obtained the fitness at every MCS; however, we sampled GRNs at every 10 MCS to reduce correlations. We employed the Wang-Landau method [39, 40] to determine the weights. We divided the range [0, 1] of the fitness $f$ into 100 bins. To check the consistency of the result, we performed 5 independent runs of entropic sampling. Thus 5000 GRNs on average were obtained as a total for each bin.

## Results

We performed computations on networks of $N = 16 \sim 32$ and $C = 5$ and 6. The following presents the results for $C = 5$ unless otherwise stated.

### Fitness landscape

Fig 2 shows the relative number of states $\Omega(f)$ of GRNs in each bin [$f, f + 0.01$) for both $C = 5$ and 6. The sum of $\Omega(f)$ is normalized to unity. Although this figure does not represent a conventional fitness landscape in which the fitness is drawn in the genotypic space, we may denote it as a "fitness landscape" in the same sense as the energy landscape in the protein folding problem, in which the entropy is drawn against the energy. Since the vertical axis is in the logarithmic scale, it can be interpreted as the entropy of each bin of the fitness. A majority of GRNs have a small $f$; over 97% of GRNs are within the range $f < 0.2$ for $N = 32$, $C = 5$. The fitness landscape bends at $f \simeq 0.2$ and the GRNs become exponentially rare as $f$ increases. The slopes are different for different $N$. For $f > 0.8$, the GRNs become rare with faster than exponential decay. The appearance probabilities of the GRNs participating in "the fittest ensemble", $f \in [0.99, 1]$, for $N = 32$ are as small as about $3 \times 10^{-19}$ and $1.4 \times 10^{-17}$ for $C = 5$ and 6, respectively.

### Emergence of cooperative bistability

Although the function we required for GRNs was to discriminate $I = 0$ and 1 sharply, to investigate the dynamical properties of highly-fit GRNs, we studied how the steady-state response changed at intermediate values of input. The response $\bar{x}_{out}$ (the fixed-point values) against $I$ of 12 randomly chosen GRNs with the initial state set as $x_i(0) = 0.5$ are plotted for the bin $f \in [0.7, 0.71)$ (Fig 3a) and the fittest ensemble (Fig 3b). Since the fitness expresses the difference of the response for $I = 0$ and 1, the response can either be an increasing function or a decreasing function of $I$. In the cases of $f \in [0.7, 0.71)$, most of the GRNs responded smoothly to the changes in input; some of them were ultrasensitive [41, 42], and a few among the ultrasensitive GRNs exhibited discontinuous responses. In contrast, all GRNs in the fittest ensemble exhibited discontinuous step-like responses.

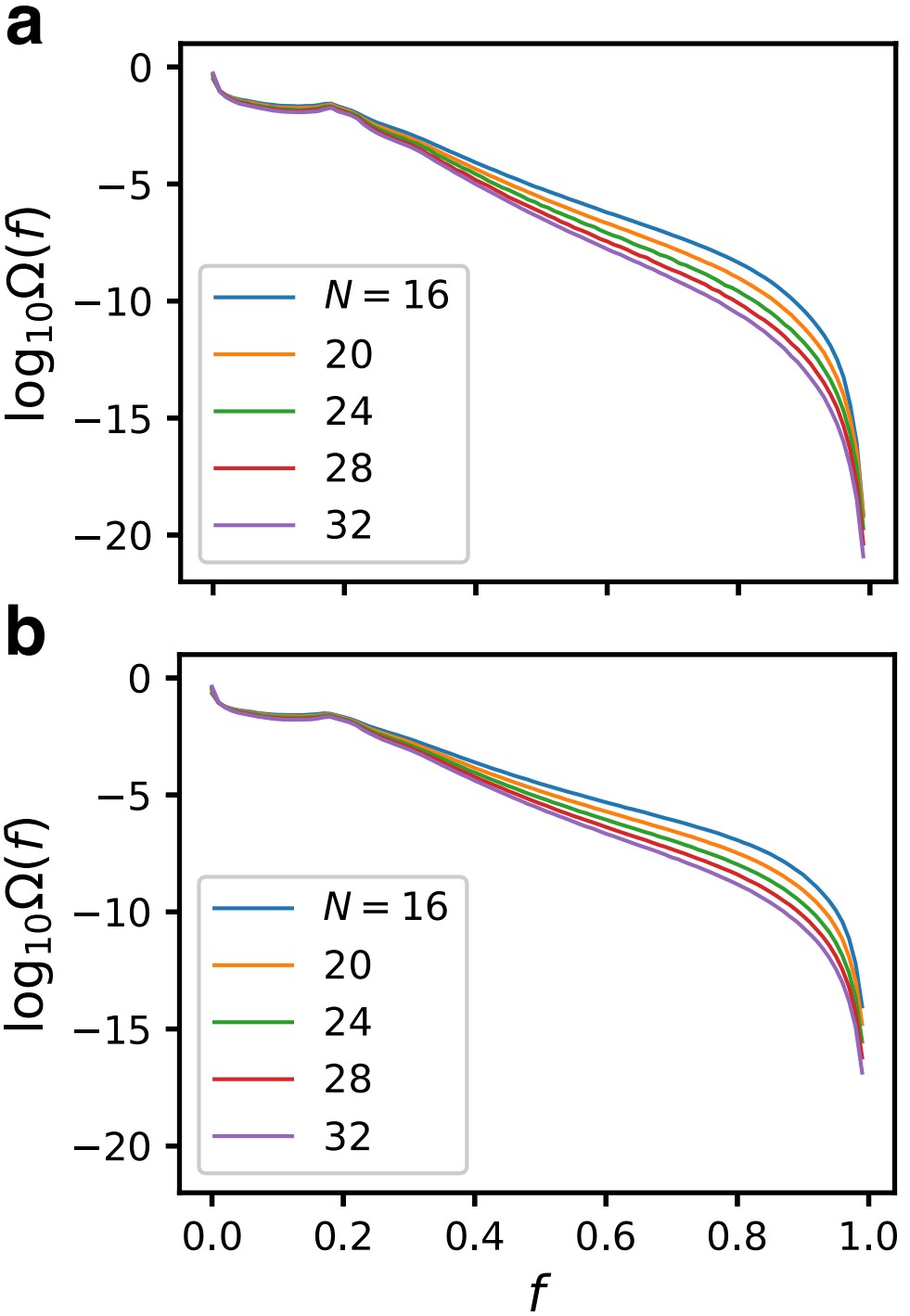

**Fig 2. "Fitness landscape", that is, the relative number of GRNs against fitness.** (a) $C = 5$ (b) $C = 6$. The vertical axes are in log scale. The whole range of the fitness is divided into 100 bins. The sum of $\Omega(f)$ for all the bins is normalized to unity. The appearance probabilities of the GRNs participating in the fittest ensemble, $f \in [0.99, 1]$, for $N = 32$ are as small as about $3 \times 10^{-19}$ and $1.4 \times 10^{-17}$ for $C = 5$ and 6, respectively.

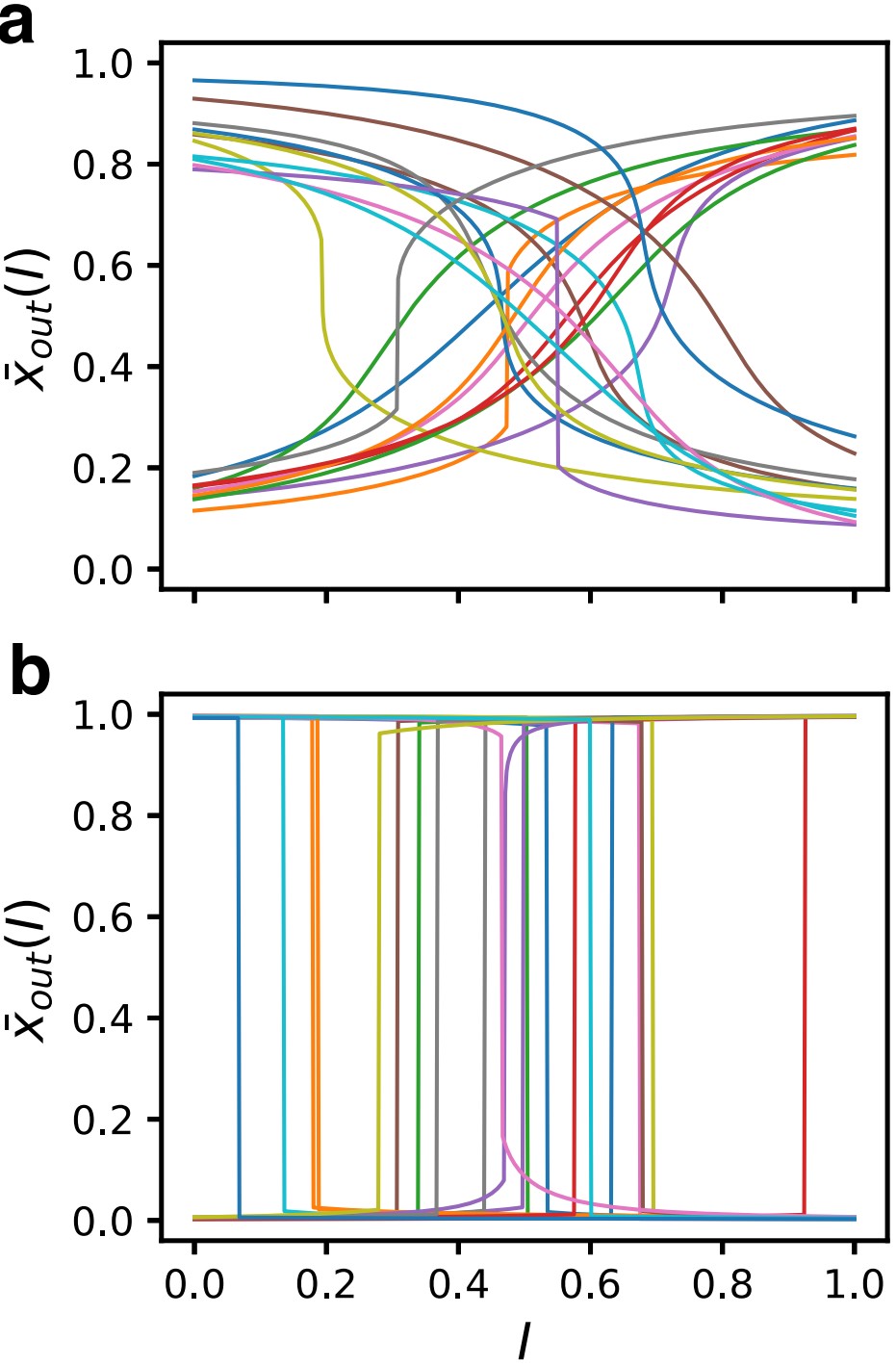

**Fig 3. Steady-state responses of GRNs for 12 randomly selected samples.** For each value of $I$, all expressions were set as $x_i = 0.5$ in the initial state and the fixed-point values are plotted. (a) $f \in [0.7, 0.71)$. Most of the samples respond smoothly to input. (b) The fittest ensemble. All GRNs exhibit a step-like discontinuous response.

These step-like responses are the consequence of two successive saddle-node bifurcations, with $I$ as the bifurcation parameter [43]. Namely, there are two stable fixed points and one unstable fixed point in a finite range of $I$. Although it is difficult to identify the unstable fixed point, there should be a bistable region between the two bifurcation points, and hysteresis is

expected to be observed between the increasing and decreasing process of $I$. We examined the hysteresis of GRN as follows: First, taking the steady-state at $I = 0$ as the initial state, we increased $I$ by 0.001 and ran the dynamics until the steady state was reached. This procedure was repeated up to $I = 1$. Next, we performed the inverse process, starting from the steady state at $I = 1$ and then decreased $I$ to $I = 0$. An example of a GRN belonging to the fittest ensemble is shown in Fig 4a, which exhibits a clear hysteresis. We also checked GRNs that did not show a discontinuous response in the ensemble for $f \in [0.7, 0.71)$ and confirmed that they did not exhibit hysteresis.

The bistable GRNs can be classified into three classes according to the range where the bistability occurs. The first is the toggle switch. For the GRNs belonging to this class, the GRNs are monostable both at $I = 0$ and 1 and the bistable range lies between them. The second is the one-way switch [43]. In this class, the bistable range includes either $I = 0$ or 1. In the third class, both $I = 0$ and 1 are included in the bistable range. Since GRNs belonging to this last class do not work as a switch, we call this class as "the unswitchable". We found that among 4524 bistable GRNs, the ratio of the toggle switch, the one-way switch, and the unswitchable were 27.8%, 42.5%, and 29.7%, respectively. The GRNs belonging to the one-way switch and the unswitchable were expected not to follow the abrupt change of the input, at least in one direction. By running the dynamics setting $I$ as 0 for the first 1000 steps, then 1 for the next 1000 steps, and 0 again for the last 1000 steps, we confirmed that the toggle switch GRNs followed the input change, while other GRNs were unable to follow the input change properly due to them being trapped by the wrong fixed point.

We then studied the fitness dependence of the proportion $P_2$ of GRNs exhibiting bistability. Since it is difficult to rigorously examine the bistability of dynamical systems with large degrees of freedom, we employed a heuristic method; if a difference larger than 0.01 was observed in the steady-state value of some $I$ between the increasing process and decreasing process with $I$ changed at the interval 0.001, it was regarded as the indicator of bistability. Since a very weak bistability may have been missed by this criterion, the obtained $P_2$ was regarded as the lower limit. We included the one-way switch and the unswitchable in the bistable cases. Fig 4b shows $P_2$ against $f$. $P_2$ exhibits a sigmoidal increase, which, for the fittest ensemble of $N = 32$, reached 99.9% (4524 among 4528 for $C = 5$ and 4785 among 4787 for $C = 6$). Since we did not observe a significant size dependence, it was not considered to be a phase transition. However, there was a characteristic value of $f$ where the bistable GRNs started to appear. From its tendency to increase, we expect $P_2$ to approach 1 as $f \rightarrow 1$. This means that the GRNs necessarily become bistable as the fitness increases. The qualitatively same result was also obtained for $C = 6$.

Although we did not require the ability to follow the input in the definition of fitness, it may be added as a requirement *a posteriori*. For example, the fitness for the GRNs which did not follow the input change may be defined as 0. Then, the fittest ensemble will be reduced in size to 28%. Using our method, the fitness can be modified even after the computation has been done. From herein, we consider the fittest ensemble, which includes GRNs that could not follow the change of input.

## Robustness against noise

In the following, we discuss several kinds of robustness for all 4528 GRNs in the fittest ensemble. First, the robustness against the input noise was considered. We assumed that the number fluctuation of the input molecules was the source of the noise. We observed an instantaneous response for the change of the input value $I = 0 \rightarrow 1 \rightarrow 0$ as previously. However, this time, the uniform random number in the range $[-0.3, 0.3]$ was added to the input as the noise. Although

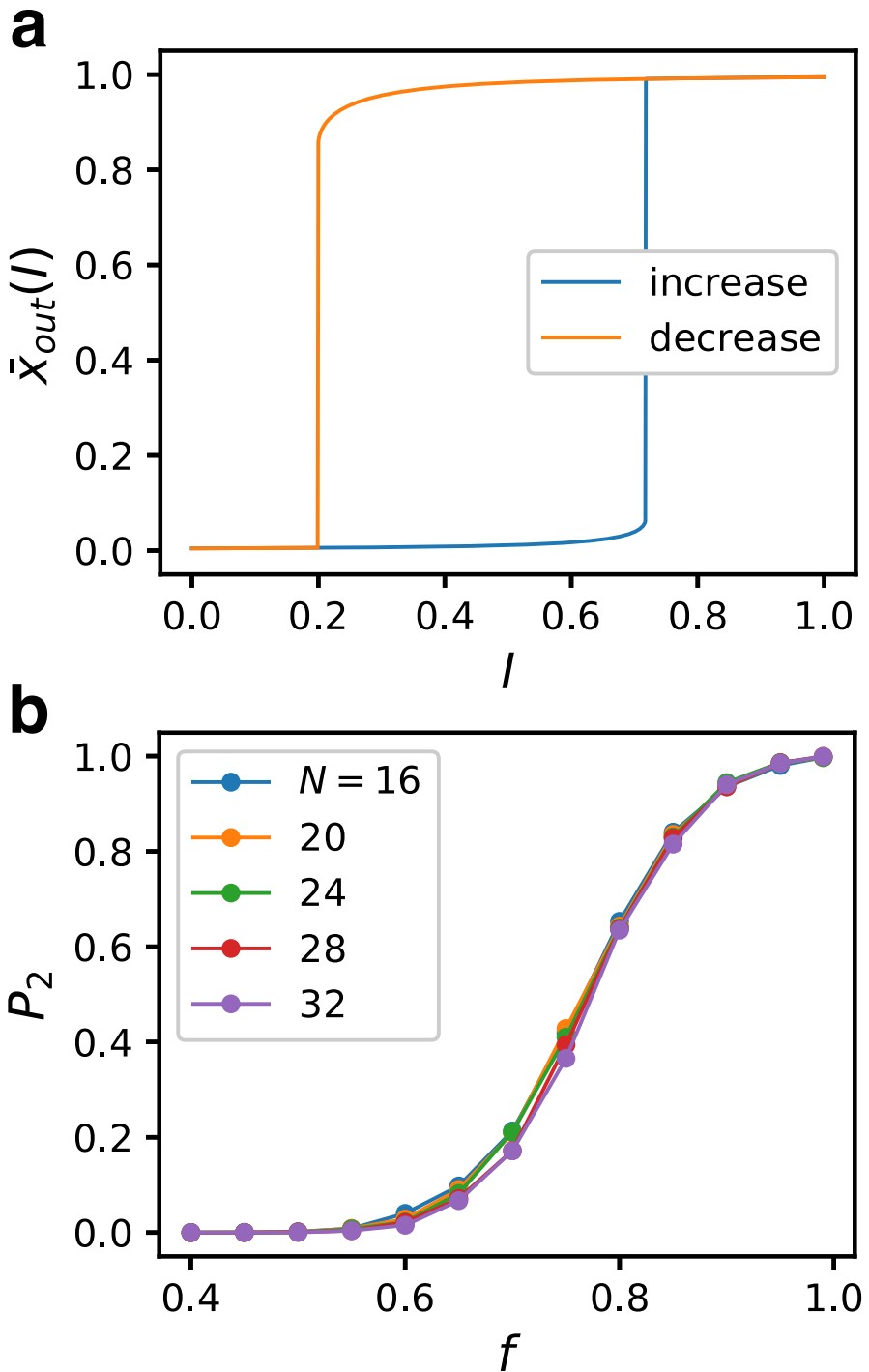

**Fig 4. Emergence of bistable responses.** (a) An example of the response of the output gene when the input was changed gradually for a GRN in the fittest ensemble for $N = 32$ and $C = 5$. The final state of the expression for $I$ was used for the initial state for the next value of $I$. The fixed-point values are plotted for both the increasing process (blue) and the decreasing process (orange) of $I$. Clear hysteresis is seen in an intermediate range of $I$. (b) Appearance probability of the bistable GRNs against $f$ for 13 bins for $C = 5$.

*I* became negative from time to time throughout this procedure, this was unlikely to cause a problem for investigating the effect of the noise qualitatively.

Fig 5a shows the dynamical response to the noisy input of the GRN which followed the input change without noise. Both the input and the response are plotted. The response was found to be stable despite the noisy input. This is the consequence of the bistability and the GRN works as a low-pass filter. We confirmed that all GRNs that followed the input properly in the absence of noise were able to also follow the noisy input. Moreover, the number of the GRNs able to follow the input increased, reaching 1506, under this condition. An example is shown in Fig 5b, which suggests that the GRN locked to the wrong fixed point was released by the noise. We call this effect the "noise-induced response" (NIR).

Next, we investigated the robustness against internal noise. Here, the number fluctuation of transcription factors was assumed to be the source of the noise. The dynamics were modified as follows:

$$x_i(t+1) = R\left( I(t)\delta_{i,1} + \sum_{j\neq i} J_{ij}\{x_j(t) + \xi_{ij}(t)\} \right), \tag{5}$$

where $\xi_{ij}(t)$ is the internal noise added to the expression of $j$-th gene when regulating $i$-th gene. The uniform random numbers in the range $[-0.1, 0.1]$ were used as $\xi_{ij}(t)$.

The temporal responses to the changes in input for both the cases with and without the internal noise are plotted in Fig 6. Fig 6a shows the same GRN as Fig 5a, which followed the input change both in the absence and the presence of noise. In Fig 6b, a GRN which did not follow the input in the absence of noise is shown. It successfully followed the input when the internal noise was applied, despite being slightly noisy. Namely, the NIR by the internal noise was observed in this case. We found that the number of GRNs with the ability to follow the input increased to 46% (2086 out of 4528) under this condition. The ratio depended on the amplitude of the noise. When a larger noise $\xi_{ij} \in [-0.2, 0.2]$ was applied, although the NIR ratio increased, some GRNs which had been able to follow the input without noise lost that ability.

The robustness against both the input noise and the internal noise are consequences of bistability. Despite the fact that the robustness was not required as the fitness, bistable GRNs appeared as the fitness increased, and, as a byproduct, they acquired the robustness against noise automatically. Therefore, the robustness against noise is an accompanying property of high fitness.

## Robustness against mutation

Finally, we investigated the robustness against mutation. We considered the effect of the simplest mutation, *i.e.*, the deletion of one of the edges. This mutation represented a situation where the affinity between a gene and a TF was lowered by a slight mutation occurring at the TF or the TF binding site.

We computed the fitness $f'$ after all possible mutations for all GRNs in the fittest ensemble. For each GRN, the input gene and the output gene were kept the same as those before the mutation. The color map in Fig 7a shows the logarithm of the probability distribution $P(f')$ against $f$. The sum of $P(f')$ in each bin of $f$ is normalized as unity. Most of $f'$'s did not differ largely from the original $f$s. Thus, the fitness did not change much after the single-edge deletion. However, the edges for which the fitness dropped significantly when deleted started to appear near $f \simeq 0.6$. As $f$ increased, the low peak corresponding to such edges (seen as light blue area) moved to $f' \simeq 0$ and the edges with intermediate $f'$ decreased. Fig 7b shows the distribution of $f'$ for the fittest ensemble. The edges were divided into two groups: edges for

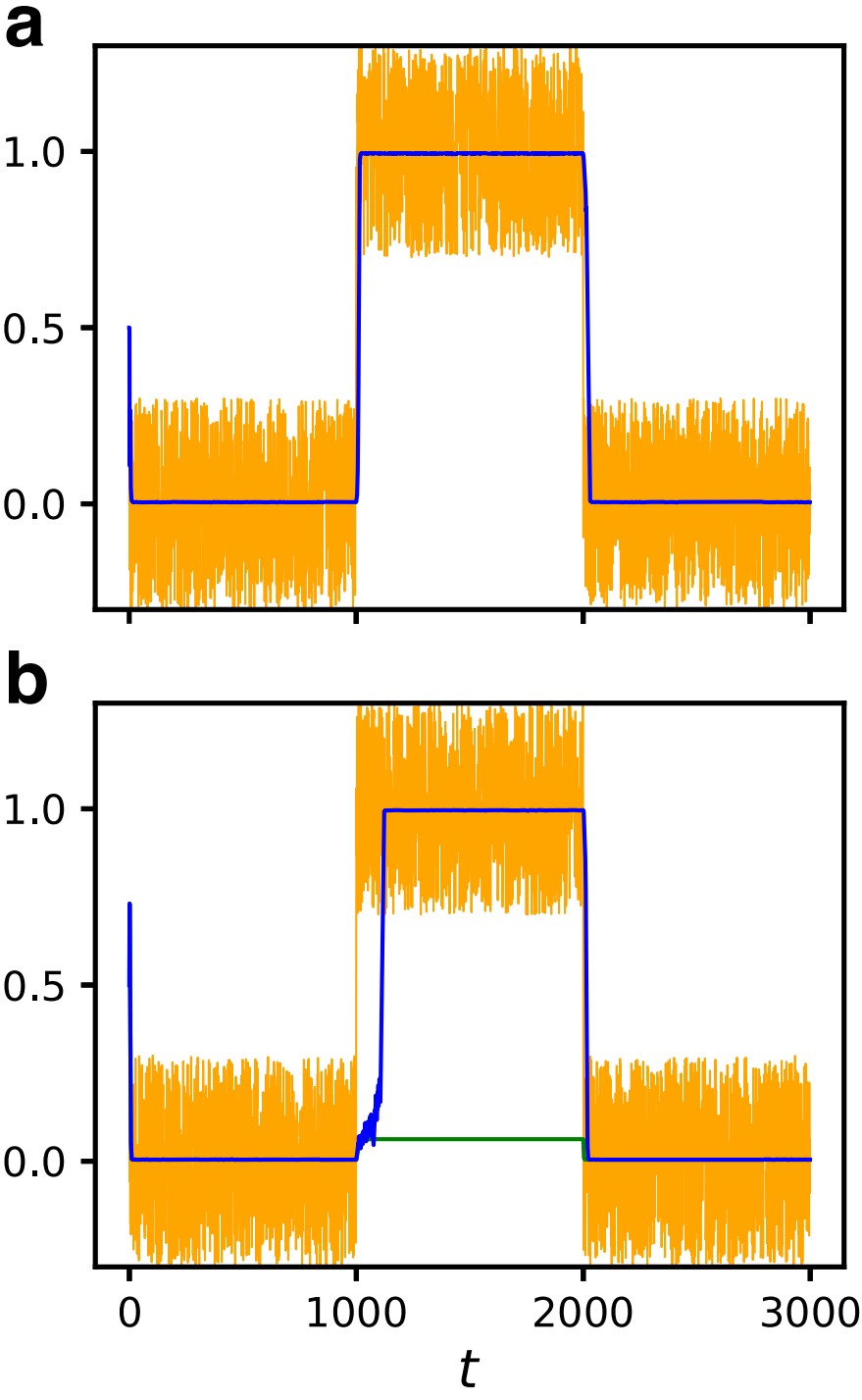

**Fig 5. Dynamical responses to noisy input for $N = 32$ and $C = 5$.** Both the input $I$ (orange) and the response of the output gene $x_{out}(t)$ (blue) are shown. The input value was changed as $I = 0 \rightarrow 1 \rightarrow 0$ at every 1000 steps with uniform random number $\in [-0.3, 0.3]$ being added at every time step. The green line indicates the response without noise. (a) The case that the GRN can follow the input change without noise. (b) The case that the GRN fails to follow the input change without noise. The noise-induced response (NIR) is observed.

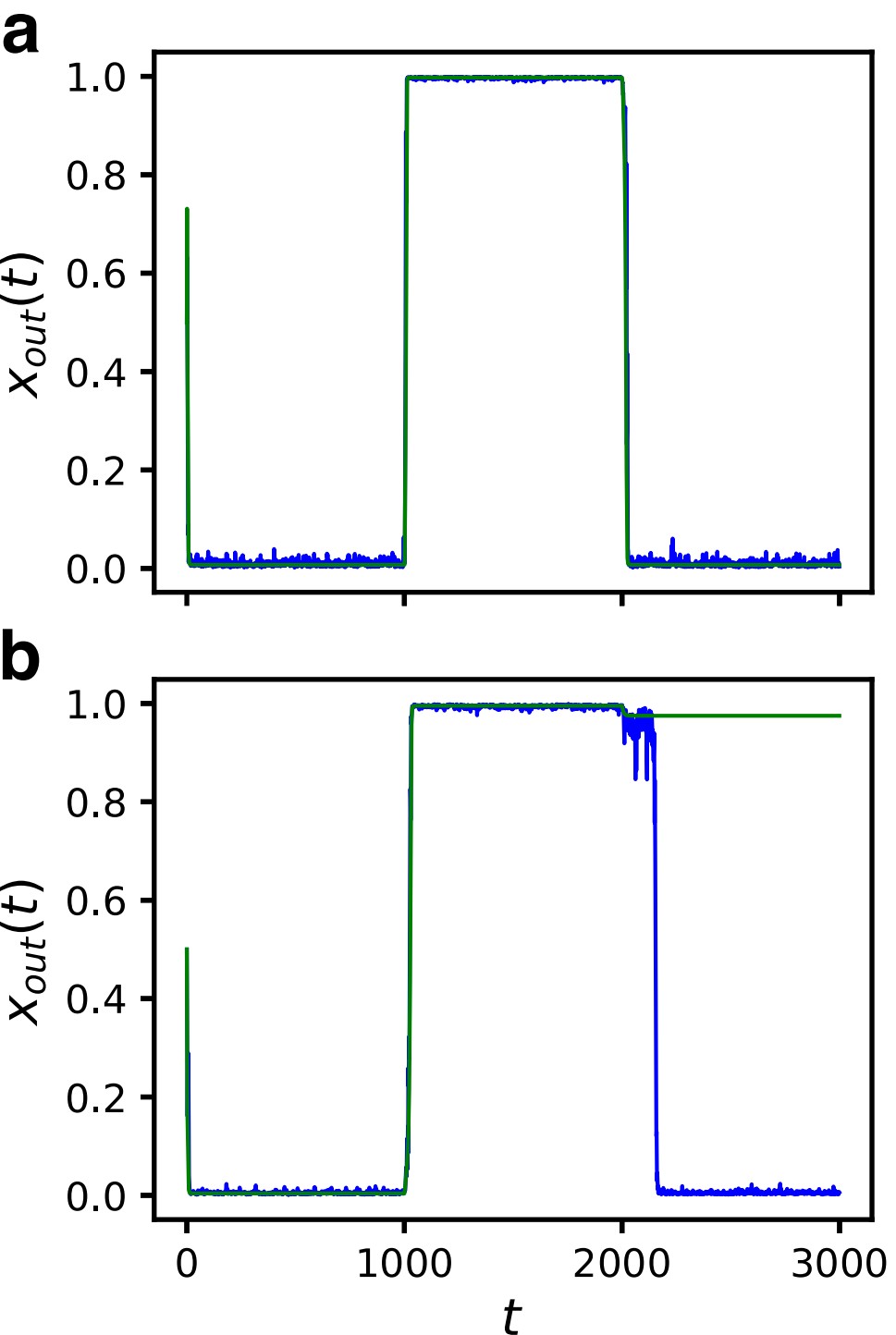

**Fig 6. Dynamical responses under internal noise for $N$ = 32 and $C$ = 5.** The input value was changed as $I = 0 \rightarrow 1 \rightarrow 0$ at every 1000 steps with uniform random number $\in [-0.1, 0.1]$ being added to the input to each gene at every time step. The responses of the output gene $x_{out}$ with (orange) and without (green) internal noise are shown. (a) The same sample as Fig 5a. (b) A case where the GRN fails to follow the input change without noise but exhibits NIR.

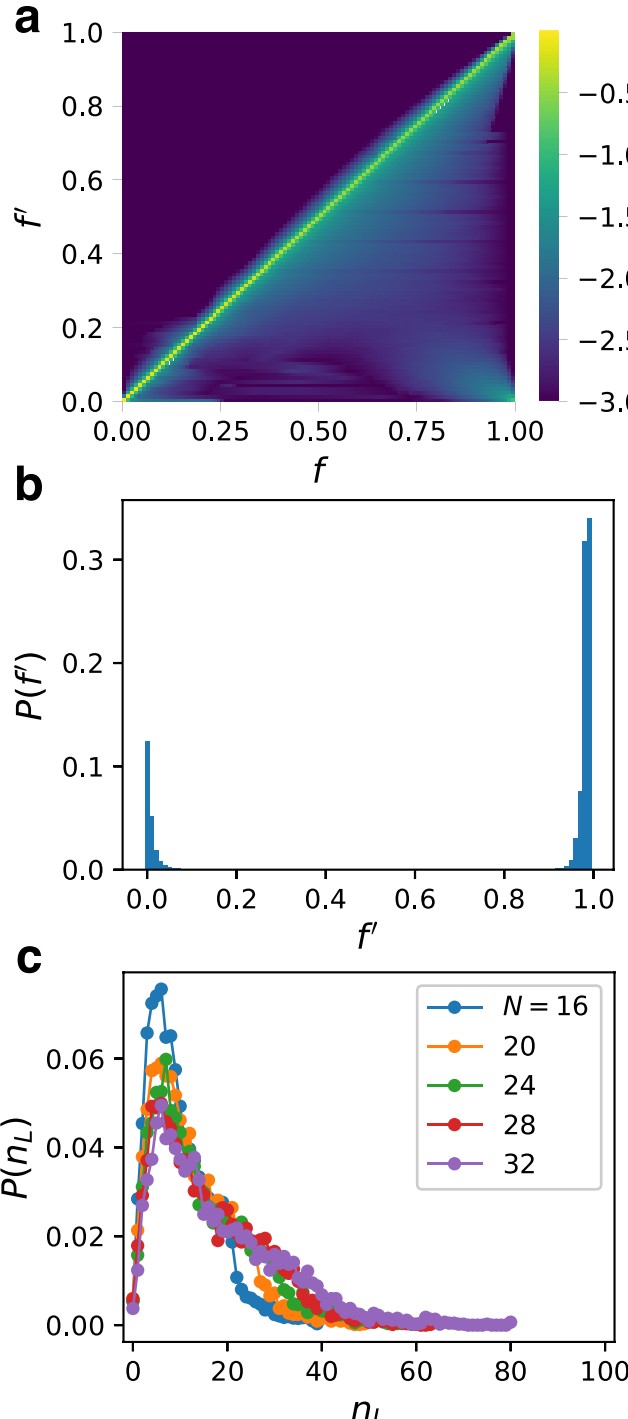

**Fig 7. Robustness against mutation.** (a) The probability distribution $P(f')$ of the fitness $f'$ after all possible single-edge deletions against $f$ for $N = 32$. The values of $f'$ are divided into 100 bins as $f$. Sum of $P(f')$ for each bin of $f$ is normalized to unity. The bins for $P(f') < 0.001$ are shown in the same color. (b) The probability distribution $P(f')$ for the fittest ensemble for $N = 32$. (c) The probability distribution $P(n_L)$ of the number of the lethal edges $n_L$ for the fittest ensemble.

which $f'$ stayed high and edges with $f'$ near zero. Although there were a few intermediate edges, they were scarce, and are not visible in the figure. In other words, as the fitness increased, the lethal edges started to appear and the edges were divided into neutral ones and lethal ones. The majority of the edges were neutral against mutations and the lethal edges were less common, even in the fittest ensemble.

We counted the number of lethal edges $n_L$ for each GRN in the fittest ensemble. Fig 7c shows the probability distribution $P(n_L)$. Here, a large threshold was set, where $f' < 0.9$ was regarded as the criterion for the lethal edges. However, since most of the edges were either $f' \simeq 1$ or $f' \simeq 0$, the choice of threshold only slightly affected the result. Although there was a size effect in $P(n_L)$, interestingly, the peak position did not depend on size significantly. The typical numbers of the lethal edges were about 6 and 7 for $C = 5$ and 6, respectively. This implies that large GRNs readily become relatively more robust than smaller ones. There were completely robust GRNs without a lethal edge, although they were scarce. For $N = 32$, the number of such GRNs was 17 out of 4528, which is a small number, however, they were not extremely rare considering the rareness of the fittest ensemble.

Among these 17 GRNs, 13 were toggle switches and the rest four were one-way switches. This ratio of the toggle switches is significantly higher than that for all the GRNs in the fittest ensemble in S1 Fig. We show the network structures of all 13 such GRNs in S1 Fig. We call them "toggle switches without a lethal edge" (TSwoLE). For TSwoLE, we studied the effects of another type of mutation, namely, the addition of a single edge. We tried all the possible single-edge additions for each GRN. We found that all the GRNs kept high fitness for more than 95% of possible single-edge additions. In contrast, this ratio differs largely from network to network for other GRNs belonging to the fittest ensemble. Thus, TSwoLE are particularly robust to mutation.

To induce a stronger mutation, we performed a single-node deletion, that is, a knockout of a single gene to all the GRNs in the fittest ensemble. Again, most of the genes were divided into neutral genes and lethal genes for the fittest ensemble. The probability distributions of the number of lethal nodes are shown in Fig 8. Since the effect of the mutation was strong, we did not find GRNs without lethal nodes. However, relatively robust GRNs with only a few lethal nodes were not extremely rare.

## Motif analysis

Network motifs are defined as "patterns of interconnections that recur in many different parts of a network at frequencies much higher than those found in randomized networks" [27, 44]. We investigated the distributions of motifs consisting of three nodes and three edges. Since self-regulation and the mutual regulation are prohibited in our model, three nodes and three edges form a triangular loop. Such loops are classified into two classes: the feedback loops (FBL) and the feedforward loops (FFL). FBL is classified further into the positive FBL (+FBL) and the negative FBL (-FBL). The former includes an even number of repressions and the latter includes an odd number of repressions. Similarly, FFL is classified further into the coherent FFL (+FFL), which includes an even number of repressions, and the incoherent FFL (-FFL), which includes an odd number of repressions.

For the GRNs in the fittest ensemble for $N = 32$ and $C = 5$, we counted the number of these loops. Fig 9a shows the number distributions of the four types of loops. As a reference, we also show the results for $f \in [0, 0.01]$ (Fig 9b), which can be regarded as the random ensemble of GRNs. For the random ensemble, the distributions of +FBL and -FBL agreed with each other as expected and those for +FFL and -FFL agreed with each other as well. The fittest ensemble exhibited a different tendency. The loop that was observed most frequently was +FFL, and the

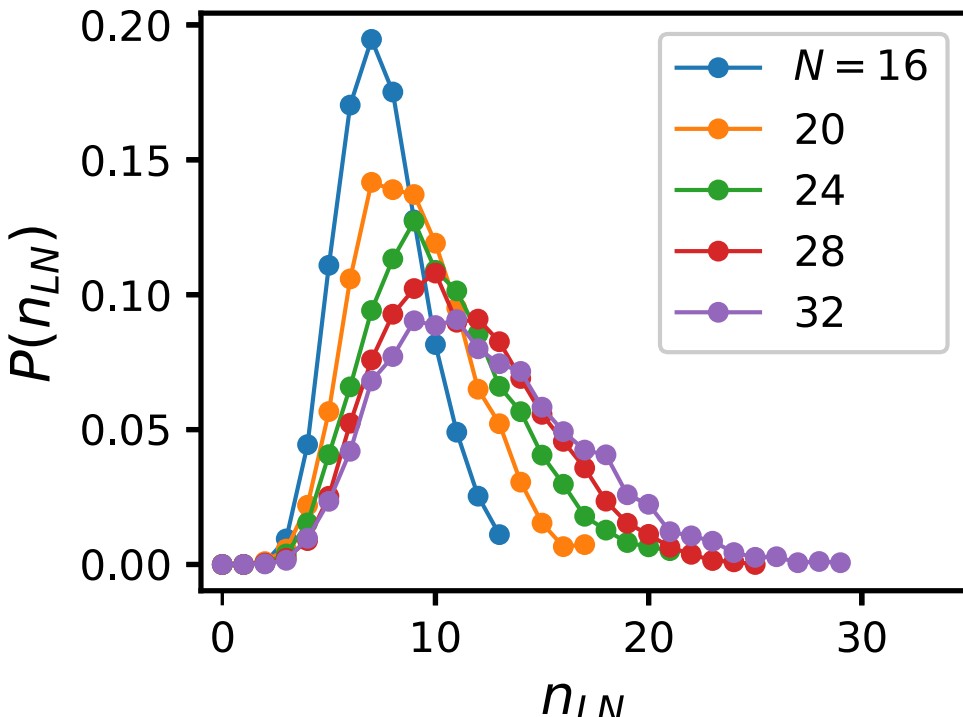

**Fig 8. Probability distribution $P(n_{LN})$ of the number of lethal nodes $n_{LN}$ for the fittest ensemble.**

next was +FBL. These two types of loops were significantly abundant compared to the random ensemble and thus were considered to be motifs. In contrast, -FBL and -FFL were scarce compared to the random ensemble.

## Discussion

We carried out a multicanonical Monte Carlo computation to sample random gene regulatory networks (GRNs). By classifying them according to the fitness, we investigated the fitness-dependent properties of these GRNs. For high fitness to be realized, an ultrasensitive response is needed for GRNs and the result strongly suggests that all GRNs with the maximum fitness in our model exhibit bistability. This bistability is considered to be a cooperative phenomenon of many genes. In this study, we defined fitness as the difference between the steady-state responses to two different inputs, "on" and "off", when started from the same initial state. Since we did not assume the bistability explicitly in fitness, the bistability is an emerging property. We found three different categories of GRNs among the bistable GRNs: the toggle switch, the one-way switch, and the unswitchable. They are considered to play different roles if realized in biological systems. We can suppress the appearance of the unswitchable, which may not have a biological role, by changing the definition of fitness. However, we expect that the bistability appears even with such fitness, as long as we require a large difference in the response between two input states.

Bistability and hysteresis are widely observed in living systems, such as in the *lac* operon, the family of MAPK cascades, and the bacteriophage λ, and have been extensively studied both experimentally and theoretically [25, 26, 43, 45–71]. There are many examples of toggle switches and one-way switches. One of the well-known toggle switches is found in the lysogenic-lytic transition of the phage λ [25, 26, 65, 72]. The cdk1 activation system of the Xenopus

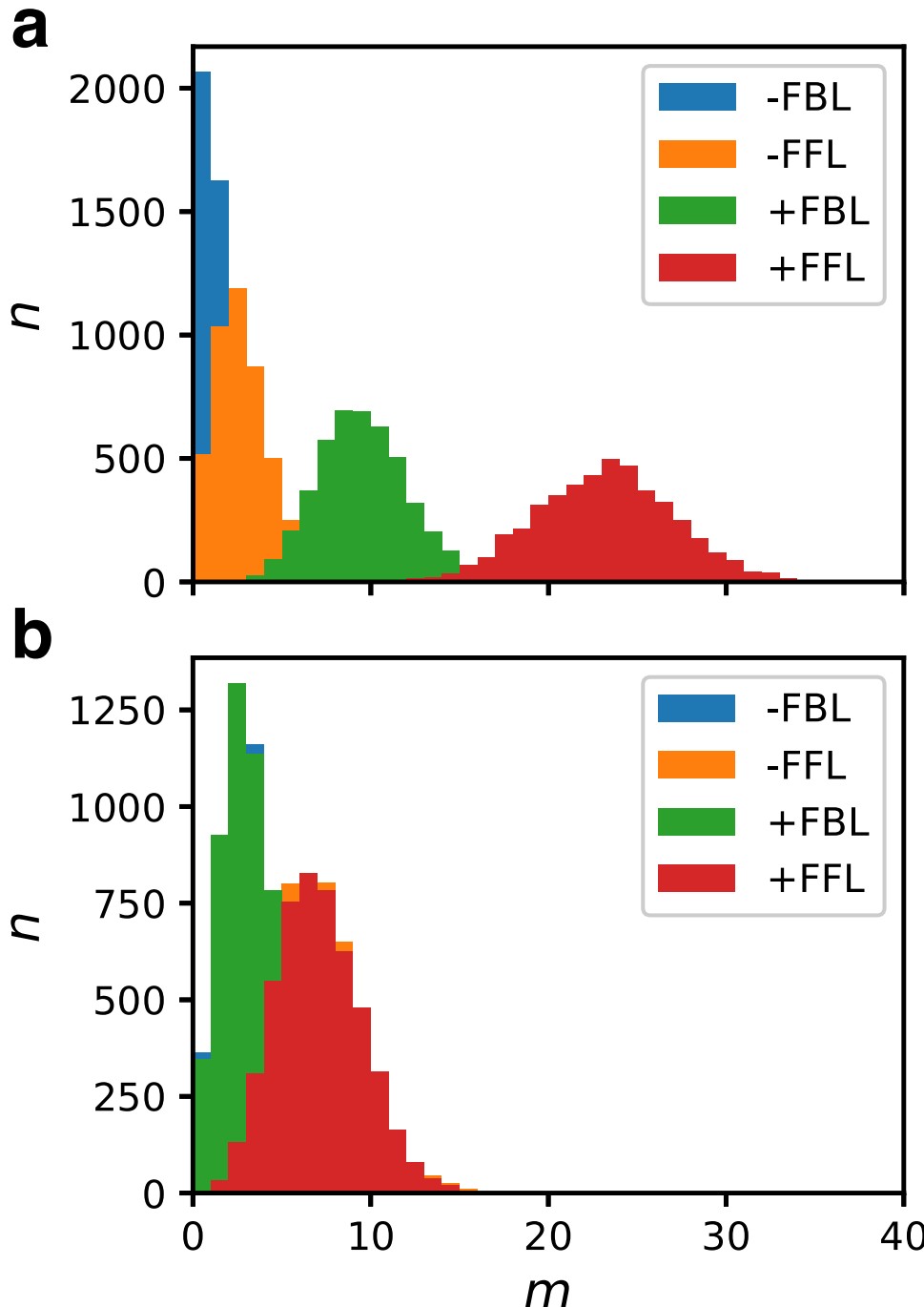

**Fig 9. Number distribution of the triangular loops.** (a) The GRNs in the fittest ensemble for $N = 32$ and $C = 5$. The distribution of the positive feedback loop (+FBL), the negative feedback loop (-FBL), the coherent feedforward loop (+FFL) and the incoherent feedforward loop (-FFL) are shown. (b) The GRNs for $f \in [0, 0.01]$, which can be regarded as the ensemble of random GRNs.

egg is another example, which behaves as a reversible bistable switch having a clear hysteresis in response to cyclin B1 concentration [53, 56, 68]. One-way switches are utilized in GRNs related to cell-fate decisions. A widely-analyzed example is the maturation of the Xenopus oocyte, which is regulated by the bistability of the MAPK cascade [47]. However, a relatively

small number of components were considered for discussing the mechanism of bistability in most cases. In contrast, the bistable GRNs that we found in this study consisted of a large number of genes. The present result indicates a possibility that more genes than considered participate to realize bistability in biological systems.

We analyzed the network motifs consisting of three edges for GRNs in the fittest ensemble. We found that the motif with the largest abundance was the coherent feedforward loop (+FFL) and the next was the positive feedback loop (+FBL). In real GRNs, FFLs are the most frequently observed motifs [27]. Shen-Orr *et al.* reported that +FFL is a characteristic motif in the GRN of *E. Coli* [44]. Later studies revealed that the incoherent feedforward loop (-FFL) is the next abundant motif [73]. Similar observations were also obtained for yeast [27, 74]. The result of the present study agrees with these observations in part because +FFL was the most abundant. In contrast, -FFL was rather scarce in our model. +FFL is known to generate a sign-sensitive delay [27]. However, since we considered only the steady-state response in the definition of fitness, such delay may not be important. To explain +FFL abundance, we assumed that it is utilized for creating a large response from a "sloppy" response function [11].

The second abundant motif, +FBL, generates bistability when combined with ultrasensitive responses [42, 47, 51, 57, 69, 70, 75, 76]. Typical structures are the double-negative FBL and the double-positive FBL. Burda *et al.* investigated the motifs of functional GRNs by randomly generating GRNs using the Markov chain Monte Carlo method [22]. They found that a combination of double-negative FBL and self-activation is ubiquitous in GRNs exhibiting multistability. Such a structure is found in phage λ [25, 26] and is utilized for realizing bistability. This structure works as a bistable switch rather robustly, even when the kinetic parameters are randomly changed [77]. It is also known to have a potential for exhibiting tristability, observed, for example, in the GATA1–PU.1 system [78, 79]. In contrast, since mutual regulation and self-regulation are prohibited in our model, such a structure is not possible. However, the +FBL abundance suggests that +FBLs are needed to exhibit bistability. We consider that structures playing equivalent roles to the double-positive FBL are formed by several combinations of +FBLs. Conversely, since some GRNs in the fittest ensemble lack -FFL and/or -FBL, they are not mandatory for bistability. Our result that particular motifs overexpress as the fitness becomes high is consistent with the finding by Burda *et al.*, wherein a motif appropriate for function emerges automatically. For understanding the roles of motifs, we require a more detailed analysis based on a larger number of samples.

Inoue and Kaneko [11] argued that a large number of genes need to work cooperatively to obtain a reliable response, notably, in cases where the genes are "sloppy". Although they did not report bistability, our results are consistent with their observation, since the cooperative bistability in the present study was found to be a cause of a reliable response.

The importance of noise, which originated from the finiteness of the number of molecules such as transcription factors, in the gene regulation systems has been emphasized by previous studies [8, 11, 36, 61, 63, 64, 80]. We found that, although robustness against noise was not taken into account in the definition of fitness, robustness against both the input noise and the internal noise was acquired automatically as a byproduct of bistability. The cause and effect could be reversed; if we required robustness against noise as fitness in evolutionary simulations, bistable GRNs would evolve because bistable GRNs are highly robust. We also found noise-induced response (NIR) to a change in the input. Some studies attributed the origin of the bimodal distribution of the cell states to the switching of the bistable systems induced by noise [48, 50, 63, 64, 67]. Our NIR systems should also show the bimodal distribution after the environmental state changes if many identical GRNs are considered.

The emergence of new fixed points can be considered an "innovation" [81] or "a big evolutionary jump". Since cooperative bistability and robustness against noise are the consequence

of high fitness, we conclude that this "evolutionary jump" occurs inevitably as the fitness increases irrespective of the evolutionary pathway. This can be denoted as "the universality" of evolution. In other words, the possible phenotypes are restricted by the fitness function and GRNs with two stable fixed points necessarily appear through evolution. If evolution was rewound and repeated over and over again, the evolving genotypes would be different each time. However, the phenotypes would have the above properties in common, as long as the same fitness function is used. This can also be interpreted as a possible mechanism of convergent evolution.

As for robustness against mutation, we found that the regulating interactions are divided into two categories: neutral and lethal. The lethal interactions were comparatively scarce. Similar results were obtained for the genes. Although a direct comparison is difficult because the context is different, there is evidence from a comprehensive single-gene knockout experiment that lethal genes are indeed scarce [82]. These lethal genes are considered to be essential for the function of GRNs. Interestingly, we found a few GRNs that had no lethal interactions. In such cases, the function is realized cooperatively by all the interactions. We also found that larger GRNs become relatively more robust than the smaller ones.

Isaran *et al.* reported a rewiring experiment for GRN of *E. Coli*, namely, they added new regulatory interactions to GRNs [83]. They reported that the cells were robust for most cases of rewiring. To compare with this experiment, we made comprehensive single-edge addition to the obtained GRNs classified as the "toggle switches without a lethal edge" (TSwoLE) and confirmed that they are robust for most of such mutations. That is, these GRNs are robust both for single-edge deletion and single-edge addition. This result is consistent with the rewiring experiment. Although such GRNs are a minority among the fittest ensemble, they are not so scarce. Robustness for edge addition is advantageous in acquiring new functions and is considered as a possible source of evolvability.

Ciliberti *et al.* conducted a numerical experiment in which GRNs were sampled randomly [19]. They found that the functional GRNs formed a large cluster in the neutral space. Although exploring the structure of the neutral space from our result is not straightforward, the fittest ensemble is close in concept to the neutral space, and the fact that many edges are neutral for deletion indicates that the robust GRNs are not isolated in the fittest ensemble.

What can be said about the mutational robustness from this study is that robust GRNs are not extremely rare among the highly-fit GRNs. In other words, a high fitness is not necessarily accompanied by fragility. Even if evolution was a simple optimization process, there would be some chance for robust GRNs to evolve. However, the evolutionary process is far from the random-sampling process used in this study. Rather, mutational robustness is considered to be enhanced during evolution. Even so, the fact that the robust GRNs are not extremely rare is important because the destination of evolution can be chosen only from the repertoires available in the set of possible GRNs. The present result implies that the robust GRNs are readily obtained through evolution.

There has been some numerical evidence of a correlation between mutational robustness and noise robustness [8, 19, 36]. However, in the present study, while the robustness against noise was a direct consequence of bistability, the relationship between bistability and mutational robustness was not clear. In this context, a suggestive finding is that most of the GRNs in TSwoLE are toggle switches. This is in contrast to other GRNs in the fittest ensemble, in which only 27.8% are toggle switches. This point should be explored further and will be investigated in a future study.

Finally, we remark on the methodology. The rare event sampling method that we used can be readily extended to the multidimensional landscapes that the evolutionary pathway goes

through, and will be useful to explore the structure of the neutral space. A comparison with evolutionary simulations is ongoing.

## Supporting information

**S1 Fig. All GRNs classified as "toggle switch without lethal edge".** All the GRNs which act as toggle switches and have no lethal edge in the fittest ensemble are shown. Blue lines indicate activation, and red lines indicate repression. The interaction matrices for these GRNs are available at Zenodo (DOI: 10.5281/zenodo.3716026).
(PDF)

## Acknowledgments

The authors thank Koich Fujimoto, Masayo Inoue, Kunihiko Kaneko, Katsuyoshi Matsushita, Tomoyuki Obuchi, Nen Saito, Hiroki Sayama, Kei Tokita, and Hajime Yoshino for their fruitful discussions and comments.

## Author Contributions

**Conceptualization:** Shintaro Nagata, Macoto Kikuchi.

**Funding acquisition:** Macoto Kikuchi.

**Investigation:** Shintaro Nagata, Macoto Kikuchi.

**Methodology:** Shintaro Nagata, Macoto Kikuchi.

**Project administration:** Macoto Kikuchi.

**Software:** Shintaro Nagata, Macoto Kikuchi.

**Writing – original draft:** Macoto Kikuchi.

**Writing – review & editing:** Shintaro Nagata, Macoto Kikuchi.

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
