## [Decision Letter · Decision Letter 0]

4 Dec 2019

Dear Dr Kikuchi,

Thank you very much for submitting your manuscript 'Emergence of  cooperative bistability and robustness of gene regulatory networks' for review by PLOS Computational Biology. Your manuscript has been fully evaluated by the PLOS Computational Biology editorial team and in this case also by independent peer reviewers. We apologize for the delay in the review process. The reviewers appreciated the attention to the problem of studying robustness in gene regulatory networks, but raised several concerns about the manuscript as it currently stands. While your manuscript cannot be accepted in its present form, we are willing to consider a revised version in which the issues raised by the reviewers have been adequately addressed. We cannot, of course, promise publication at that time.

We would very much like to see the authors address the major and minor concerns raised by the reviewers. We would also require that the code and scripts associated with this work be made available online for others to use.

Sincerely,

Sushmita Roy, Ph.D.

Associate Editor

PLOS Computational Biology

Erik van Nimwegen

Deputy Editor

PLOS Computational Biology

[LINK]

Reviewer's Responses to Questions

**Comments to the Authors:**

Reviewer #1: In this manuscript, the authors apply MC sampling to GRN fitness landscape. The model structure, its rationale, the methodology to sample models, and the results, are all well described. Some interesting types of behaviours of GRNs are displayed, including the toggle-switch-response, and noise induced response. This type of work helps improve our understanding and is an important complementary type of work compared with simulations of specific models for GRNs. My comments are only minor:

1) Code used for the sampling and model simulations seems not to be available. It would be useful to do so.

2) Similarly for specific examples of GRNs with e.g. high fitness.

3) It would useful if a bit more specific connections could be made to experimental evidence for some of the properties that are discussed. I understand results with these simple networks cannot be directly tested but for some of the suggestions that can be based on this work, data might be available in other labs/literature.

4) Related to this, it should also be possible to make an explicit comparison between the networks studied here, and available known GRNs from various organisms. Can anything be said about how similar/dissimilar the most fit networks in this study, and GRNs from various organisms, are?

5) Similarly, a bit more explicit comparison with literature would be warranted. Other studies where fitness landscapes of GRN are sampled or somehow analyzed (without applying evolutionary approaches) are mostly not used.

6) Is it possible to further analyze the selected GRNS (e.g. the most fit ones) to further get insight into properties of such 'best networks'? E.g. does the presence/absence of certain network sub-motifs correlate with fitness or with other properties of interest?

7) page 7 mentions "An inverse-λ distribution". It is not clear to me what this means.

Reviewer #2: In the present manuscript, Nagata and Kikuchi explored the fitness landscape of gene regulatory networks (GRNs) and analyzed the emergence of robustness. They considered a toy model of GRNs with one input gene and one output gene, and utilized the multicanonical Monte Carlo method to classify the GRNs per their fitness. They showed that GRNs can acquire bistability to distinguish between the two different input states – ‘ON’ and ‘OFF’ as the fitness increases. As the bistable GRNs are robust again noise, the robustness of GRNs can be regarded as a byproduct of high fitness. Overall, the study has been well designed and performed.

Regarding the robustness of GRNs, there is a hypothesis that the dynamics behaviors of GRNs are mainly determined by the topologies of the GRNs. The following relevant papers should be appropriately cited and discussed.

Cell. 2009 Aug 21;138(4):760-73. doi: 10.1016/j.cell.2009.06.013.

PLoS Comput Biol. 2017 Mar 31;13(3):e1005456. doi: 10.1371/journal.pcbi.1005456

BMC Syst Biol. 2018 Jun 19;12(1):74. doi: 10.1186/s12918-018-0594-6.

Regarding the bistability or multi-stability of GRNs, the authors may want to discuss more biological examples, such as the genetic toggle switch GATA-1/PU.1 governing lineage differentiation. The following relevant studies should be appropriately cited and discussed.

Dev Biol. 2007 May 15;305(2):695-713.

Phys Biol. 2017 May 23;14(3):035007. doi: 10.1088/1478-3975/aa6f90.

Reviewer #3: This interesting manuscript presents a prototypical model of a gene regulatory network that responds to a signal. A node of this network is designated as output and the difference between the expression of this node in the two extreme values of the signal (0 and 1) is defined as the network’s fitness. The manuscript shows that the overwhelming majority of the networks in the highest fitness category are bistable (they have two fixed points for certain values of the signal). Moreover, these high-fitness networks are robust to noise, and even exhibit noise-induced response, i.e. the presence of noise in the perceived value of the signal or of internal nodes helped the system respond to a 0-1-0 pattern in the signal. These networks also exhibit significant resilience to mutations in edges or nodes. The manuscript suggests that robustness against noise or mutation is a byproduct of high fitness, and may be readily acquired through evolution.

I think these suggestions are intriguing and the manuscript will be a valuable contribution to the literature if a number of unclear points are addressed.

1. The introduction starts with presenting “function” as a foundational concept and characteristic of living systems. It is at this point that the embodiment of function assumed in this work needs to be clearly presented and justified. There also should be some (later) discussion on the dependence of the conclusions on the assumed embodiment of function. For example, the signal is assumed to be continuous and in the range [0,1], yet the assumed function depends on the difference in response to the two extreme values. The bistable, high fitness networks have low sensitivity to smaller changes in the signal. Is that acceptable biologically?

2. The abstract refers to “switching the fixed points” and emergence of new phenotypes. Even after reading the manuscript twice this is not clear to this reviewer. Probably what is needed here is an intuitive way to refer to bistability.

3. Two criteria in network construction seem to be partially contradictory. First (line 79) paths must exist from all the nodes to the output node. Second (line 126), the node that exhibited the largest sensitivity except the input node was selected as the output node. What was done if the first criterion was not satisfied for node that exhibited the largest sensitivity?

4. The decision to prohibit self-regulation and mutual regulation of two genes should be motivated a bit more than “for the sake of simplicity” and “do not cause any problem”. Perhaps the reason was a wish to avoid known/expected sources of bistability so that the results uncover less known or unexpected sources.

5. The explanation for choosing a tanh function instead of a Hill function (in line 104) is not sufficient. Perhaps the reason has to do with having to choose a Hill coefficient value in the latter case? What would change if a Hill function with relatively large Hill coefficient were used?

6. The sentence in lines 110- 113 seems important but it is not clear. Why do the levels of expression decay from one gene to the next? What exactly do you mean by feed-forward type regulations? Maybe you mean a feed-forward loop, which is the meeting of shorter and longer paths in the same target node?

7. In lines 236-238, what is the intuitive meaning of the case that the region of bistability includes I=0, I=1, or both? Which of the three cases is referred to as one-way switch?

More minor points:

1. Please check the description that introduces gene regulatory networks to avoid the misinterpretation that genes interact with each other or regulate each other. It needs to be said clearly at least once that gene products, chiefly proteins, interact with each other. For example, transcription factor proteins regulate the transcription of genes into mRNA. Response to the environment that ultimately leads to gene regulation also involves protein-protein interactions, miRNA- RNA interactions and reactions. The framework can plausibly be used in all of these cases so there is no need for a restrictive “gene” language.

2. In line 43, “fitness is taken as a parameter instead of the genotype”. One would rarely consider genotype a parameter. Maybe you meant “variable”, not “parameter”, for both cases?

3. In line 57, “model we dealt with” would be better as “model we constructed”

4. In line 63, “to the direction” should be “in the direction”.

5. In line 364, “cause and result” should be “cause and effect”.

6. The manuscript occasionally uses “the” instead of “a”, or no article. “The” should only be used if it is clear from the context that there is only one of the noun that follows. For example, in line 189, “computations on the networks of N=16~32” should be “computations on networks with N=16~32”

**Have all data underlying the figures and results presented in the manuscript been provided?**

Reviewer #1: No: Some of the models (those used for the specific figures on one or two networks) should be made available.

Reviewer #2: Yes

Reviewer #3: No: No data are provided; this is in line with the traditions of reporting the results of simulation-based studies.

Providing the edge lists of the 24 networks whose steady state response is shown in Figure 3 a and b would be an appropriate and informative example of data sharing.

PLOS authors have the option to publish the peer review history of their article (what does this mean?). If published, this will include your full peer review and any attached files.

Reviewer #1: No

Reviewer #2: No

Reviewer #3: No

---

## [Decision Letter · Decision Letter 1]

19 May 2020

Dear Dr. Kikuchi,

We are pleased to inform you that your manuscript 'Emergence of  cooperative bistability and robustness of gene regulatory networks' has been provisionally accepted for publication in PLOS Computational Biology.

Best regards,

Sushmita Roy, Ph.D.

Associate Editor

PLOS Computational Biology

Erik van Nimwegen

Deputy Editor

PLOS Computational Biology

Reviewer's Responses to Questions

**Comments to the Authors:**

Reviewer #1: My previous concerns have been addressed in a satisfactory way.

Reviewer #2: The authors have appropriately addressed the referees' comments.

Reviewer #3: The revised manuscript successfully addresses all my points.

**Have all data underlying the figures and results presented in the manuscript been provided?**

Reviewer #1: Yes

Reviewer #2: Yes

Reviewer #3: Yes

PLOS authors have the option to publish the peer review history of their article (what does this mean?). If published, this will include your full peer review and any attached files.

Reviewer #1: No

Reviewer #2: No

Reviewer #3: Yes: Reka Albert

---

## [Editor Report · Acceptance letter]

23 Jun 2020

PCOMPBIOL-D-19-01701R1 

Emergence of  cooperative bistability and robustness of gene regulatory networks

Dear Dr Kikuchi,

I am pleased to inform you that your manuscript has been formally accepted for publication in PLOS Computational Biology. Your manuscript is now with our production department and you will be notified of the publication date in due course.

With kind regards,

Laura Mallard
